# Simulation of transient rolling resistance of bicycle tyres at various ambient temperatures

**Jukka Hyttinen**[1,2☯], **Malte Rothhämel** [2☯] *, **Jenny Jerrelind** [2☯], **Lars Drugge**[2☯]

**1** Scania CV AB, Södertälje, Sweden, **2** KTH Royal Institute of Technology, Engineering Mechanics, ECO[2] Vehicle Design, Stockholm, Sweden

☯ These authors contributed equally to this work.

* malter@kth.se

**Data Availability Statement:** All relevant data are within the paper and its Supporting information files.

**Funding:** The project belongs to the Centre for ECO2 Vehicle Design, which is funded by the

## Abstract

The range of an electrically assisted bicycle, which is constrained by the rider's cycling ability and the battery capacity, is heavily influenced by rolling resistance. Furthermore, the magnitude of rolling resistance affects commuters' motivation to decide whether to cycle or to choose another way to commute. This paper presents a way to simulate the transient rolling resistance of bicycle tyres as a function of ambient temperature. The significance of the change in driving resistance at different ambient temperatures is demonstrated through the range simulation of an electrically assisted bicycle at varying ambient temperatures. A representative driving cycle for bicycle commuters was created, enabling comparison of dynamic behaviour in a standardised set, to evaluate the effect of ambient temperature on the battery capacity and the increase in driving resistances. To the authors' knowledge, this kind of model has not previously been created for bicycles. The model calculates tyre temperature based on the heat transfer, considering the heating—i. e., rolling resistance—and cooling effects—i. e., convective and radiative cooling. The decrease in tyre temperature results in an increase in rolling resistance and a decrease in the battery capacity, which was considered in the simulations. The results show significantly increased energy demand at a very low ambient temperature (down to −30˚C) compared to + 20˚C. The novelty of this article is simulating energy expenditure of bicycle dynamically as a function of ambient temperature. This model includes a temperature-dependent transient bicycle rolling resistance model as well as a battery capacity model. The findings provide researchers with a better comprehension of parameters affecting energy expenditure of bicycles at different ambient or tyre temperatures. The models can be used as a tool during the design process of bicycles to quantify the required battery capacities at different climates. In addition, traffic planners can use the model to assess the effect of changes in infrastructure on motivation to utilise bicycles.

## Introduction

Cycling is healthy [1] and may contribute to reducing greenhouse gas emissions, congestion, and air pollution [2], but unfortunately, cycling is also physically demanding. Therefore, the

Swedish Innovation Agency Vinnova (Grant Number 2016-05195). The authors wish to thank the Centre for ECO2 Vehicle Design, the strategic research area TRENoP and Scania for their financial support. The funders had no role in study design, data collection and analysis, decision to publish, or preparation of the manuscript.

**Competing interests:** The authors have declared that no competing interests exist.

trip duration is an essential aspect of utility cycling motivation [3, 4]. Moreover, additional travel time when cycling is considered three times more unpleasant than additional travel time using other forms of commuting, such as walking, driving a car, or using public transport [5]. Perceived driving comfort is connected to driving resistance forces and the efficiency of the bicycle. Typically, aerodynamic, rolling, acceleration, and climbing resistance forces are named to describe the longitudinal dynamics of vehicles as bicycles. Depending on the application, riders attempt to reduce some of these driving resistances in order to reduce driving effort. For example, rolling resistance and aerodynamic forces are reduced in competitive cycling to maintain high acceleration and high speeds on a given track, including ascending slopes [6]. In contrast, utility cyclists try to avoid hills to reduce the cycling effort required [7] and are willing to pedal a longer distance instead.

Another approach to reducing cycling effort is to utilise the assistance of an electric motor. These types of bicycles are called pedelecs. Currently, the most common pedelecs have assistive torque proportional to the rider's pedalling torque, and the rider can usually manually choose this gain factor [8], typically by means of an electronic driver interface at the handlebar. The available studies regarding pedelecs focus on the health benefits gained from cycling despite the electric assistance, which reduces the required effort [8–10]. Independent of how much assistance drivers utilise individually or on average when riding their pedelec, Yang and Lee [8] reported range anxiety for riders of pedelecs, which means the fear of not arriving at the destination, before the battery is empty. This kind of human behaviour is also observed in the context of the usage of electric cars [11]. Since the range of electric vehicles depends on many parameters, such as rolling resistance, it is worth investigating the influence of rolling resistance on the driving range of electrically assisted bikes. Specifically during wintertime cycling can be a challenge, and several studies discuss different ways of enabling winter cycling [12, 13]. However, the willingness to cycle has been shown to drop during the winter season, for example, because of increased rolling resistance [14]. Electrical assistance could counteract this decreased willingness by compensating the increase in air and rolling resistance. Even if the need for electric energy increases in this case, both the energy demand and the greenhouse gas emissions are still lower than any other modes of transport that utilise some type of non-human propulsion system [15, 16]. For that reason, the present work focuses on the effect of ambient temperature on steady-state and transient rolling resistance.

Some well-known tyre parameters affecting rolling resistance are vertical load, tyre inflation pressure, and tyre width [17, 18]. In addition, the tube type and speed dependency of rolling resistance has been known for many years. Baldissera et al. [6] confirmed this increase in rolling resistance coefficient with increasing speed in their experiments. Nonetheless, this increase has minimal relevance for utility cyclists since these changes occur at velocities much higher than are typical for utility cyclists. Furthermore, the driving losses affect not only the assisted range of the batteries but also the maximum acceleration of bicycles and pedelecs [19].

In contrast, information regarding the transient rolling resistance of bicycle tyres caused by the tyre temperature changes during driving is practically non-existent. Bicycle tyres are made of filler-reinforced rubber, which is known to have a significant temperature dependency for dissipative properties. Therefore, it is surprising that this factor has not yet been studied in more detail, and the rolling resistance coefficient is often considered only as a constant value. Previously, Hyttinen et al. [20] have shown significant temperature dependency in truck tyres. Rothhämel [21] discovered a similar temperature dependency for bicycle tyres, where decreased ambient temperature caused a significant increase in steady-state rolling resistance measurements. For temperatures below freezing point the influence of temperature was higher than the influence of tyre pressure, which otherwise has been known to be the most important factor for rolling resistance. However, the transient rolling resistance behaviour of bicycle

tyres was not investigated in Rothhämel's study. In addition to tyre temperature, also road type [19], slush, snow, and wet conditions have been shown to affect tyre rolling resistance [22, 23]. Fenre and Klein-Paste [22] noted that increase in rolling resistance correlated well with subjective discomfort experienced by cyclists. Therefore, this article aims to use the results provided by Rothhämel and expand the model to consider transient rolling resistance and the warm-up behaviour of the tyre.

Electrically assisted bicycles should have a sufficient range to incentivise people to use bicycles for short-range commuting instead of using cars. When the assisted range is long enough at all operational conditions, cyclists might perceive cycling more positively, not worrying about too short of a driving range. Another sustainability perspective is that a better understanding of rolling resistance helps to dimension batteries, so pedelecs do not have too large batteries, saving valuable natural resources. Therefore, energy consumption and battery drain for a hypothetical pedelec are studied in this article using the developed simulation model, showing how the driving range decreases at different temperatures for the chosen drive cycle.

To the authors' knowledge, the driving cycle presented in this work is the first of its kind for bicycles. This driving cycle facilitates a standardised comparison of the longitudinal dynamic behaviour of bicycles. The proposed driving cycle can be employed as it is or adapted for different purposes.

This article is structured as follows: firstly, the driving losses of bicycles are introduced, then the rolling resistance simulation model is presented. The simulation results of transient rolling resistance and the driving range of an electrically assisted bicycle are presented thereafter. Finally, the findings are discussed while conclusions and future work in the last section close the article.

## Driving resistance of a bicycle

This section describes driving losses that a cyclist experiences during cycling. The total driving losses $F_{total}$, as illustrated by Fig 1, consist of four different contributions: (I) rolling resistance $F_r$, (II) aerodynamic resistance $F_{aero}$, (III) climbing resistance $F_{climb}$, and (IV) acceleration

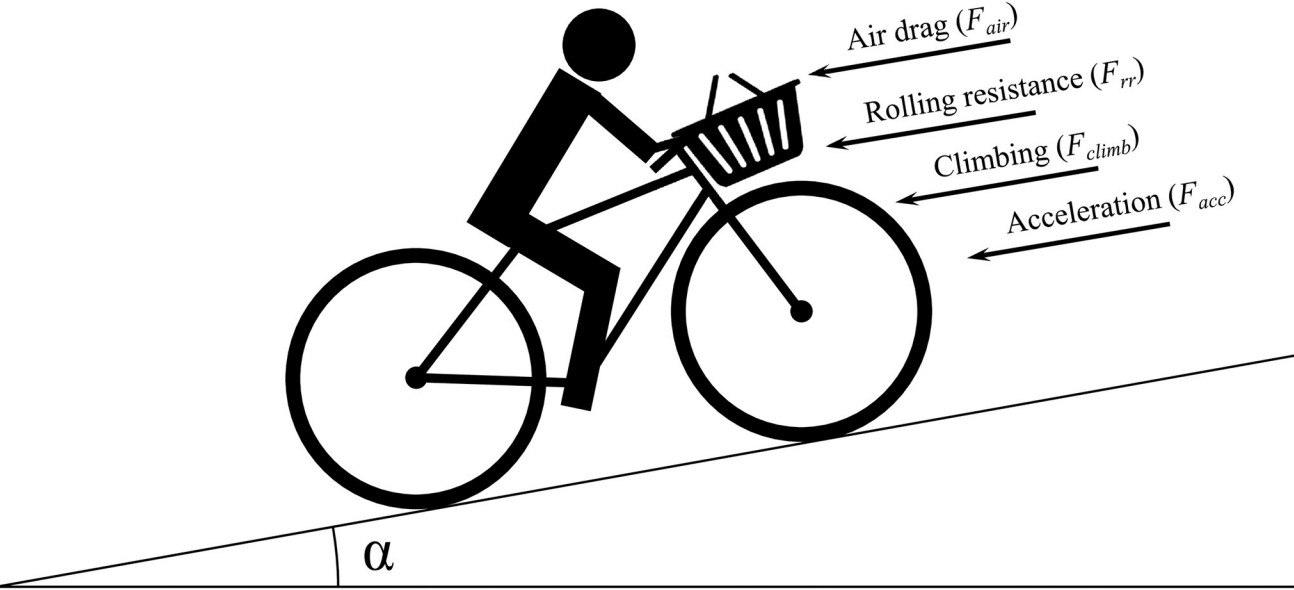

**Fig 1. Total driving losses that a cyclist must overcome.**

resistance $F_{accel}$. Eq (1) shows the calculation of the driving losses which will be explained in detail later on:

$$F_{total} = F_r + F_{aero} + F_{climb} + F_{accel} \qquad (1)$$

Rolling resistance can be explained as an energy loss related to a travelled distance with the relation $F_r = \Delta E / \Delta L$, where $\Delta E$ is the consumed energy, and $\Delta L$ is the travelled distance. This energy loss originates, for the most part, from the hysteretic behaviour of filler-reinforced rubber. Reinforcing fillers are added to rubber to make them more wear-resistant and to increase grip levels [24]. However, these reinforcing fillers increase the viscoelastic behaviour of the rubber, causing increased rolling resistance. This rolling resistance force can be represented as a rolling resistance coefficient $c_{rr}$, and it is defined as shown in Eq (2) as a ratio between the rolling resistance force and the vertical load $F_z$:

$$c_{rr} = \frac{F_r}{F_z} = \frac{F_r}{(m_b + m_c) \cdot g} \qquad (2)$$

where $m_b$ is the mass of the bicycle, including the mass of the luggage, and $m_c$ is the mass of the cyclist. Fig 2 shows how the hysteretic behaviour of rubber creates an unsymmetrical contact pressure, causing most of the rolling resistance force.

The aerodynamic resistance (Eq 3) is related to the air density $\rho_{air}$, which is a function of ambient temperature $T_a$, air drag coefficient $c_d$, the projected frontal area of bicycle and cyclist

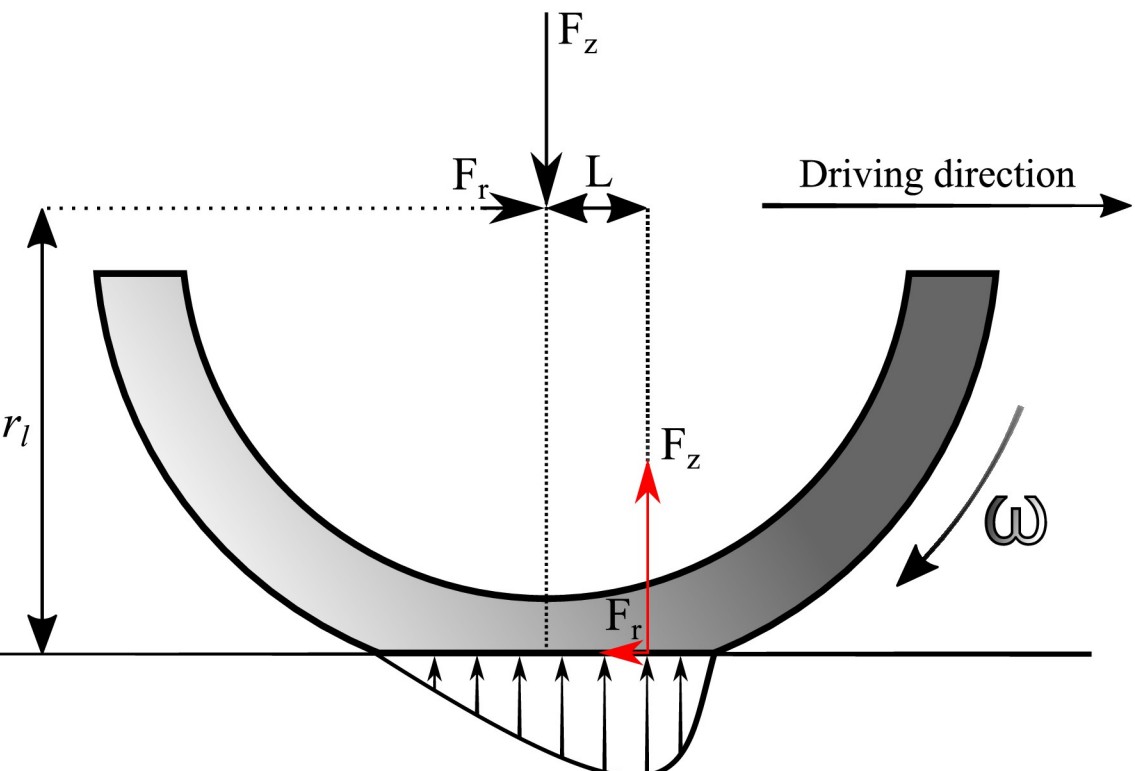

**Fig 2. The majority of rolling resistance originates from the unsymmetrical contact pressure caused by viscoelasticity in rubber.**

$A$, and vehicle speed $v$, with the following equation:

$$F_{aero}(T_a) = \frac{1}{2}\rho_{air}(T_a) \cdot c_d \cdot A \cdot v^2 \qquad (3)$$

The cross-sectional area for a typical cyclist can be assumed to be $A = 0.6$ m$^2$, and the air drag coefficient $c_d = 1.0$ [25]. At cold temperatures, the aerodynamic resistance increases because of the increase in the air density, as specified in Eq (4):

$$\rho(T_a) = \frac{p_{air}}{R_{specific} \cdot T_a} \qquad (4)$$

where $R_{specific} = 287.058\,^J/_{kgK}$ is the specific gas constant, and $p_{air}$ is the air pressure.

The inclination angle of the road $\alpha$ causes the climbing resistance (Eq (5)).

$$F_{climb} = (m_b + m_c)g \sin\alpha \qquad (5)$$

The acceleration resistance is a resistance caused by the linear inertia of the bicycle, as follows:

$$F_{accel} = (m_b + m_c)\frac{dv}{dt} \qquad (6)$$

Putting Eqs (2) to (6) together results in the following total losses

$$F_{total} = c_{rr}(T_t)m_b + \frac{1}{2}\rho_{air}(T_a) \cdot c_d \cdot A \cdot v^2 + (m_b + m_c)g \sin\alpha + (m_b + m_c)\frac{dv}{dt} \qquad (7)$$

The mass of the bicycle, luggage, and rider can be assumed to be 100 kg for a male cyclist, according to Tengattini [26]. Rotational inertia resistances also exist, but these are estimated to be negligibly low because of small rotating masses. Therefore, rotational inertia resistances are not separately calculated.

## Rolling resistance model

This section introduces the rolling resistance simulation model. The simulation model makes the following assumptions: the tyre has a uniform temperature, and the rolling resistance force is fully converted into heat. For every time step $\Delta t$, a new tyre temperature is calculated from the heat balance. The temperature-dependent rolling resistance model proposed by Rothhämel [21] is used as a basis for the transient tyre model.

The model derivation starts with a heat balance, where the net energy flow into the tyre $\dot{Q}_{net}$ is equal to the difference between in- $\dot{Q}_{in}$ and out-flowing energy $\dot{Q}_{out}$ (see Eq (8)):

$$\dot{Q}_{net} = \dot{Q}_{in} - \dot{Q}_{out} \qquad (8)$$

The energy flow stored in the tyre (Eq (9)) is described in terms of tyre mass $m_t$, specific heat capacity $c_t$ and the temperature gradient of the tyre $\frac{dT_t}{dt}$:

$$\dot{Q}_{net} = m_t c_t \frac{dT_t}{dt} \qquad (9)$$

Mandal et al. [27] previously showed that the specific heat capacity is temperature dependent. This heat capacity is modelled as a linear equation, and the values for $c_{t_v}$ and $c_{t_0}$ are

extracted from Mandal's plot for a 5 PHR nano silica-filled natural rubber:

$$c_t(T_t) = T_t c_{t_v} + c_{t_0} \tag{10}$$

The rolling resistance force causes the energy flow into the tyre:

$$\dot{Q}_{net}(v) = F_r T v = F_z c_{rr} T v \tag{11}$$

Rothhämel [21] has published Eq (12), which describes the rolling resistance at varying ambient temperatures, having two different regression parameters $b_1$ and $b_2$:

$$c_{rr}(T_t) = \frac{b_1}{T_t + b_2} \tag{12}$$

Eq (11) describes the evolution of the rolling resistance at varying tyre temperatures, and it is used in this article to describe the temperature dependency of rolling resistance. The out-flowing energy is divided into three parts: convective cooling between the wind and the tyre $\dot{Q}_{out,t-a}$, convective cooling between the road and the tyre $\dot{Q}_{out,cp}$ and radiative cooling $\dot{Q}_{out,r}$. The former is calculated according to Eq (13).

$$\dot{Q}_{out,t-a}(v) = h(v)A_t(T_t - T_a) \tag{13}$$

where $A_t$ is the outer cooling area of the tyre. The tyre is idealised to be a torus. Thereby, the outer area is calculated according to Eq (14) from the difference between the torus area $A_{torus}$ and the contact patch area $A_{cp}$:

$$A_t = A_{torus} - A_{cp} \tag{14}$$

Eq (15) gives the area of a torus:

$$A_{torus} = 4\pi^2 r_t R_t \tag{15}$$

where $R_t$ is the distance from the centre of the tube to the wheel rotational axis, and $r_t$ is the radius of the tyre tube. The contact patch area can according to Eq (16) be obtained from the inflation pressure of the tyre $p_t$ and the vertical wheel load $F_z$:

$$A_{cp} = \frac{F_z}{p_t} \tag{16}$$

The heat transfer coefficient $h$ can in accordance with Eq (17) be assumed to consist of two different contributions: free $h_{natural}$ and forced $h_{forced}$ convection, and speed-dependent convection $h_{forced}$ [28, 29]:

$$h(v) = v h_{forced} + h_{natural} \tag{17}$$

In this case, $v$ is the vehicle speed while the wind speed is assumed to be zero. The natural and forced convection parameters are extracted from Browne and Wickliffe et al. [30]. Eq (18) gives the convective cooling between the road and the tyre:

$$\dot{Q}_{out,cp}(v) = h(v)A_{cp}(T_t - T_r) \tag{18}$$

This article considers the convection coefficient equal for $\dot{Q}_{out,t-a}$ and $\dot{Q}_{out,cp}$.

$$\dot{Q}_{out,r}(v) = S e_e A_t(T_t^4 - T_r^4) \tag{19}$$

where $S = 5.67037 \cdot 10^{-8} \text{J}/_{\text{m}^2}\text{K}^4\text{s}$ is the Stefan-Boltzmann constant and $e_e = 0.96$ is the emissivity. Finally, knowing the temperature dependency of rolling resistance and cooling parameters, the temperature gradient is calculated according to:

$$\frac{dT_t}{dt} = \frac{F_r(T)v - h_t(v)A_t(T_t - T_a) - h_r(v)A_{cp}(T_t - T_r) - Se_eA_t(T_t^4 - T_a^4)}{m_t c_t} \tag{20}$$

The time integration of the tyre temperature is done using Euler integration:

$$T_{t,t+\Delta t} = T_{t,t} + \frac{dT}{dt}\Delta t \tag{21}$$

## Rolling resistance simulation

The simulation results are presented in this section. To visualise the transient rolling resistance, four consecutive 9-minute speed steps between $10 \, ^{\text{km}}/_{\text{h}}$ and $30 \, ^{\text{km}}/_{\text{h}}$ are simulated, according to Fig 3.

Figs 4 and 5 show the rolling resistance and tyre temperatures at various ambient temperatures and speed levels (Fig 3). The rolling resistance reaches different rolling resistance levels at different speed levels. The temperature increase compared to the ambient temperature becomes more prominent at higher speed levels, causing a decrease in rolling resistance. At warm temperatures, the transient rolling resistance is nearly non-existent, while it is relatively large at cold temperatures. Especially at extremely low ambient temperatures (−30˚C), the

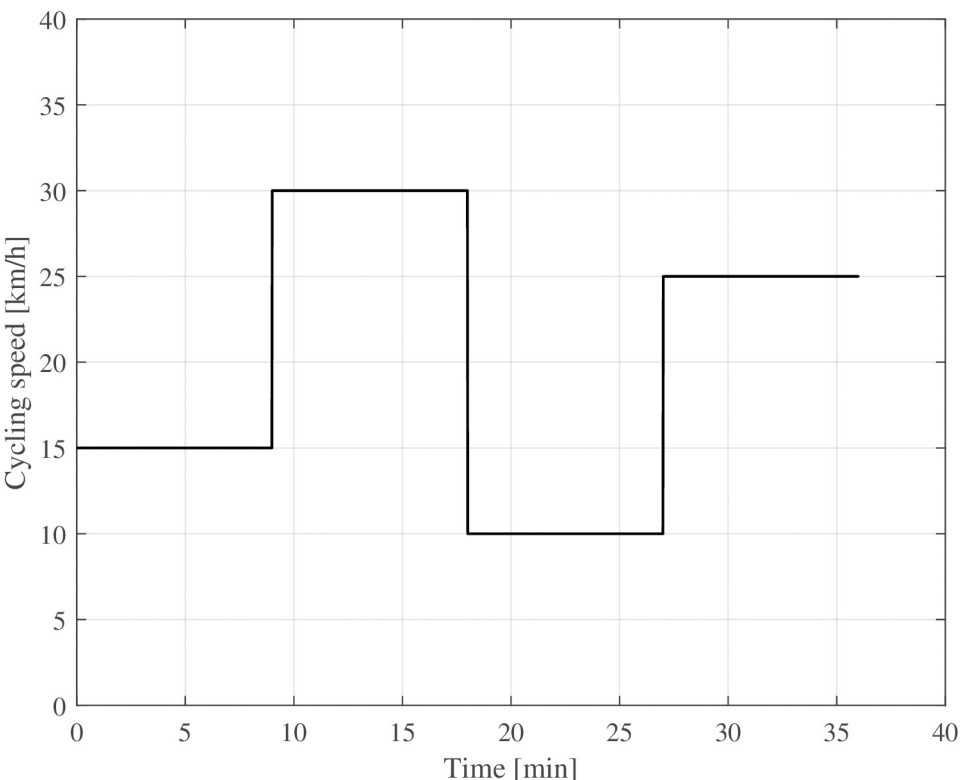

**Fig 3. Bicycle speed for steady-state simulations.**

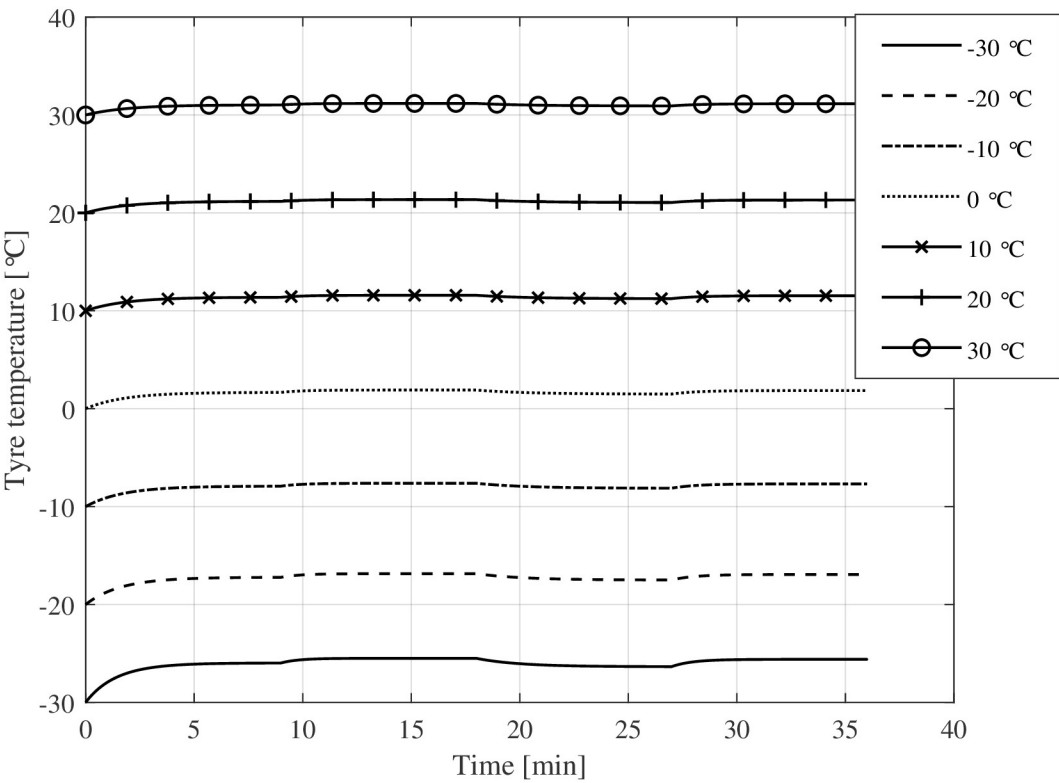

**Fig 4. Tyre temperature at varying ambient temperatures and different speed levels.**

transient rolling resistance is significant. The change is not nearly as great for bicycle tyres as it is, for example, for truck tyres [20, 31], but the change in rolling resistance is still noticeable at cold temperatures. Moreover, about 9 minutes at constant speed seems enough to reach a steady-state condition.

It has been shown previously that the motivation of cyclists decreases with increasing energy demand. This might cause cyclists to choose to commute by car instead of cycling. In addition, the motivation of cyclists can further be affected by inconvenience because of the colder ambient temperatures.

## Electrically assisted bicycles and energy consumption

The market for electrically assisted bicycles—i. e., pedelecs—is growing. For example, at the beginning of 2022 in Germany, 12% of bicycles were pedelecs [32] and 48% of all new bicycles were electrically assisted [33]. For electric motors, low ambient temperatures are usually not problematic. In contrast, batteries are known for their sensitivity to low temperatures. The most usual battery type for pedelecs is lithium-ion batteries because of their high energy density. However, the capacity of lithium-ion batteries also has a considerable temperature dependency [34]. The battery capacity decreases considerably with decreasing temperature. Eq (22) describes the change in battery capacity at different ambient temperatures:

$$B_{cap} = c + \alpha_b e^{-\beta T_b} \tag{22}$$

where $\alpha_b$, $\beta$ and $c$ are regression constants, and $T_b$ is the battery temperature in Celsius. The

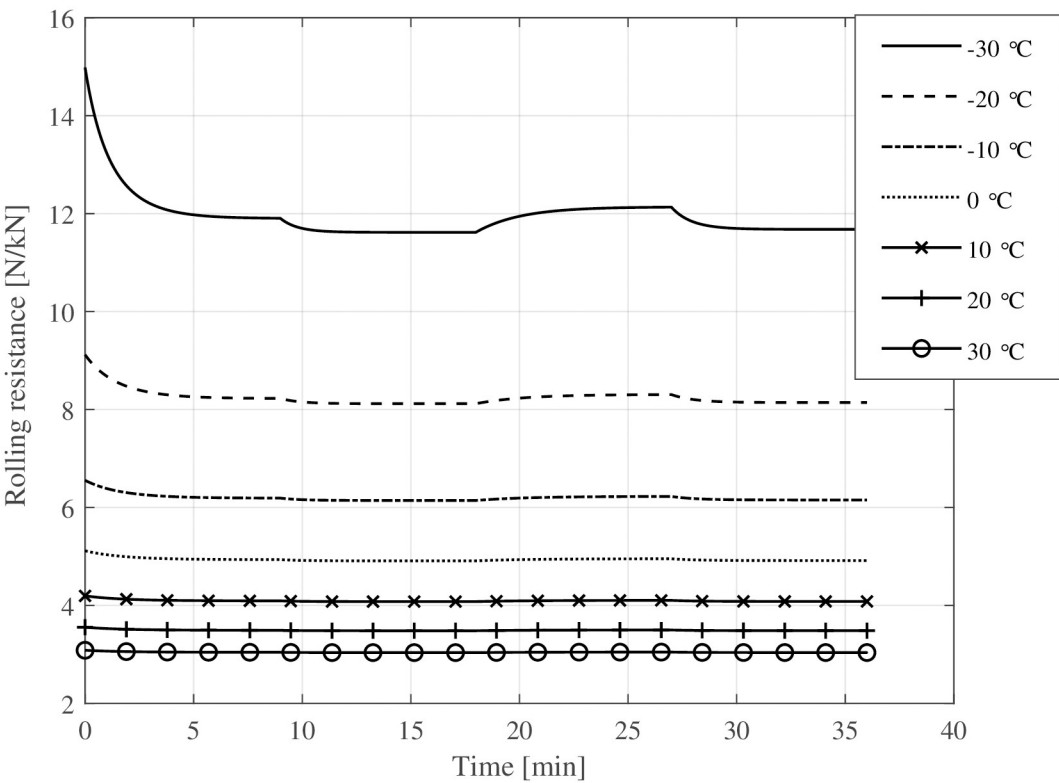

**Fig 5. Tyre rolling resistance at varying ambient temperatures and different speed levels.**

parameters for the battery capacity are taken from Zia et al. [34] $\alpha_b = -0.354$, $\beta = 0.0310$ and $c = 1.157$. At very cold temperatures ($-20°$C), the battery capacity reduces to approximately half of the capacity at $+25°$C, as shown in Fig 6. The change in battery capacity is due to changes in molecular movements. Higher temperatures increase molecular movement and enable a higher capacity than that of the nominal capacity at $+25°$C, which can lead to values above 100%. All the simulation parameters are listed in Table 1.

For the simulations, it is assumed that the pedelec goes the same driving cycle as a conventional bicycle: however, with an average speed of $21\,^{km}/_h$ in accordance with the findings by Twisk et al. [35], which is also in accordance with the experimental approach by Yang [8]. Gain factor (meaning assist level) equal to 3 was used in the simulations, which means that the electric motor assists by three times the torque that the rider is generating. So, the power of the electric motor $P_{em}$ is equal to three times the rider's power $P_{rider}$. The propulsion power is the sum of both of them. The battery capacity $W_{bat}$ is presumed to be 500 Wh.

From Schantz et al. [36], it is known that bicycle commuters cycle on average for 17 to 24 minutes. In addition, from a preliminary evaluation of measured data, it is known that bicycle commuters need to reduce their speed significantly (below 50% of the average speed) nearly once per minute. Based on this information, a driving cycle was created, as visualised in Fig 7, to study how temperature affects rolling resistance in a driving case close to reality. According to this driving cycle, the mean riding speed is $18.9\,^{km}/_h$, which is close to the data reported by Twisk et al. [35] for bicycles in the Netherlands over a 22-minute cycling period. This corresponds to a distance of 6.97 km. A cyclist and bike, with a combined mass of 100 kg, frontal

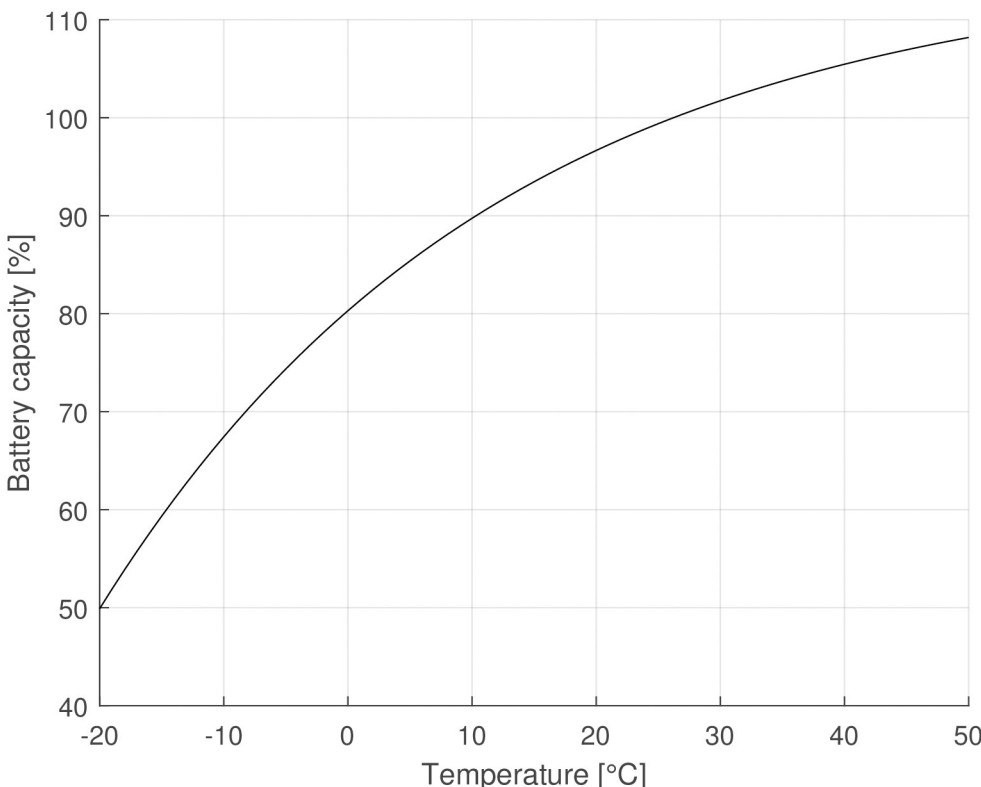

**Fig 6. Battery capacity of Li-Ion batteries over temperature, according to Zia et al. [34].** 100% corresponds to nominal capacity at + 26˚C.

**Table 1. Simulation parameters for the viscoelasticity.**

| Simulation parameters | Description |
|---|---|
| $m_t$ = 0.5 kg | Tyre mass |
| $h_{free} = 5.1 \ {}^{W}/_{m^2 K}$ | Natural convection |
| $h_{forced} = 3.59 \ {}^{Ws}/_{m^3 K}$ | Forced convection |
| $c_t = 2005 \ {}^{J}/_{kgK}$ | Heat capacity tyre |
| $w_t$ = 0.05 m | Tyre width |
| $d_{SoC}$ = 0.85 | Dynamic state of charge |
| $m_b + m_c$ = 100 kg | Total mass bicycle and cyclist |
| $e_e$ = 0.96 | Emissivity |
| $S = 5.67037 \cdot 10^{-8} \ {}^{J}/_{m^2 K^4 s}$ | Stefan-Boltzmann constant |
| $c_d$ = 1 | Aerodynamic drag coefficient |
| $W_{bat}$ = 500 Wh | Battery capacity |
| $\alpha_b$ = 0.96 | Battery capacity regression coefficient 1 |
| $\beta$ = 0.96 | Battery capacity regression coefficient 2 |
| $c$ = 0.96 | Battery capacity regression coefficient 3 |
| $b_1$ = 0.96 | Rolling resistance regression coefficient 1 |
| $b_2$ = 0.96 | Rolling resistance regression coefficient 2 |
| $A_d$ = 0.6 m$^2$ | Projected frontal area of bicycle and cyclist |
| $R_t$ = 0.28 m | Radius from wheel hub to centre of the tube |
| $r_t$ = 0.025 m | Radius of the tyre tube |

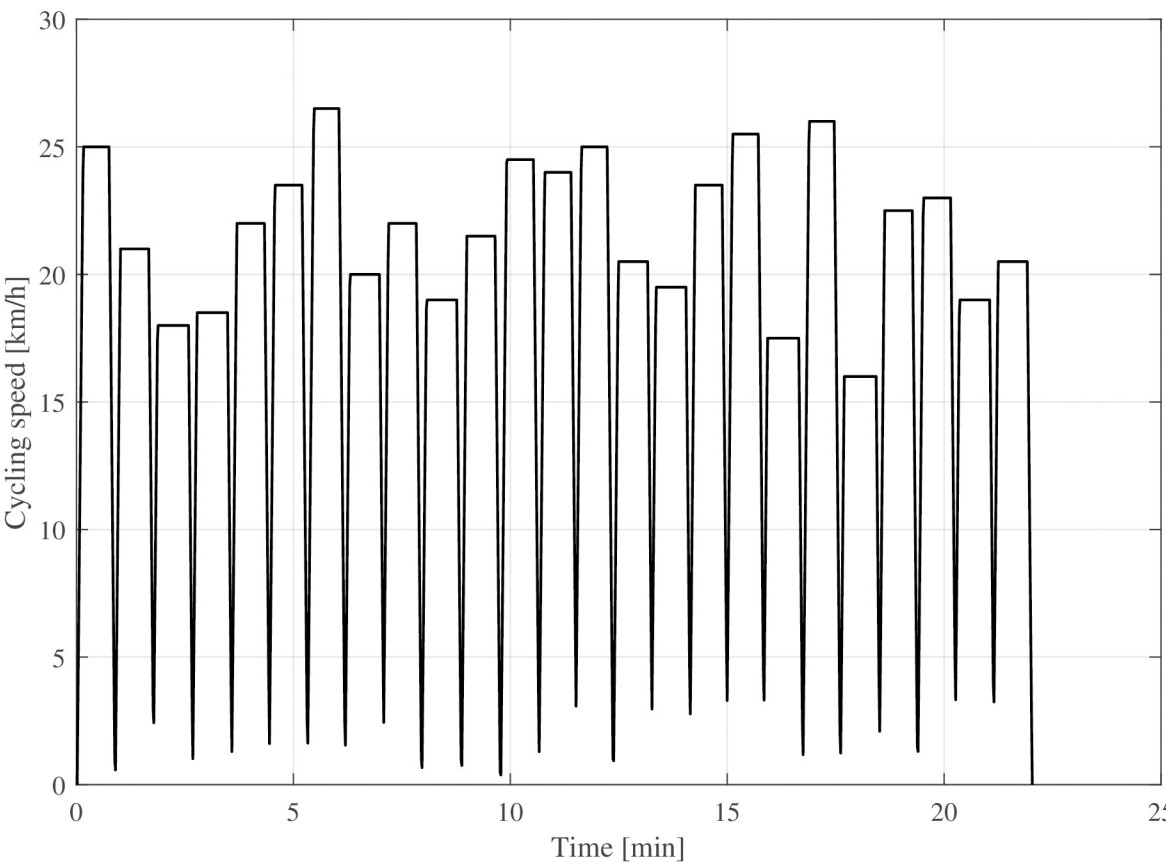

**Fig 7. The driving cycle for the range simulations.**

cross-sectional area of 0.6 m², and a drag coefficient of 1, at + 10˚C, have 170 kJ energy expenditure.

Fig 8 depicts the evolution of tyre temperature over time, converging to a nearby steady-state influenced by ambient temperature. The plot compares two scenarios where the bicycle has been stored outdoors or in a warm garage: the first where the tyre begins at ambient temperature and another where tyre temperature starts at room temperature (+ 20˚C). Additionally, Fig 9 visualises the corresponding rolling resistance due to the increasing tyre temperature. Above an ambient temperature of 0˚C, the transient rolling resistance is hardly visible when the bicycle is stored outdoors. In contrast, when the tyres start at room temperature, rolling resistance is considerably smaller below + 10˚C ambient temperatures during the first 5–10 minutes of cycling. Fig 10 shows the magnitude of the energy expenditure provided by the electrical assistance (gain factor 3), the cyclist, and the total energy expenditure.

The remaining battery capacity at different ambient temperatures is shown in Fig 11. What stands out is the significant decrease in driving range at decreasing temperatures. The battery is almost empty at a very cold temperature (−30˚C). The battery capacity would be even lower if the assist level were to be adjusted so that the cyclist experienced the same effort at all ambient temperatures.

The same driving cycle is repeated until the battery capacity is zero at all temperatures to highlight how different ambient temperatures affect the driving range. Fig 12 shows how the

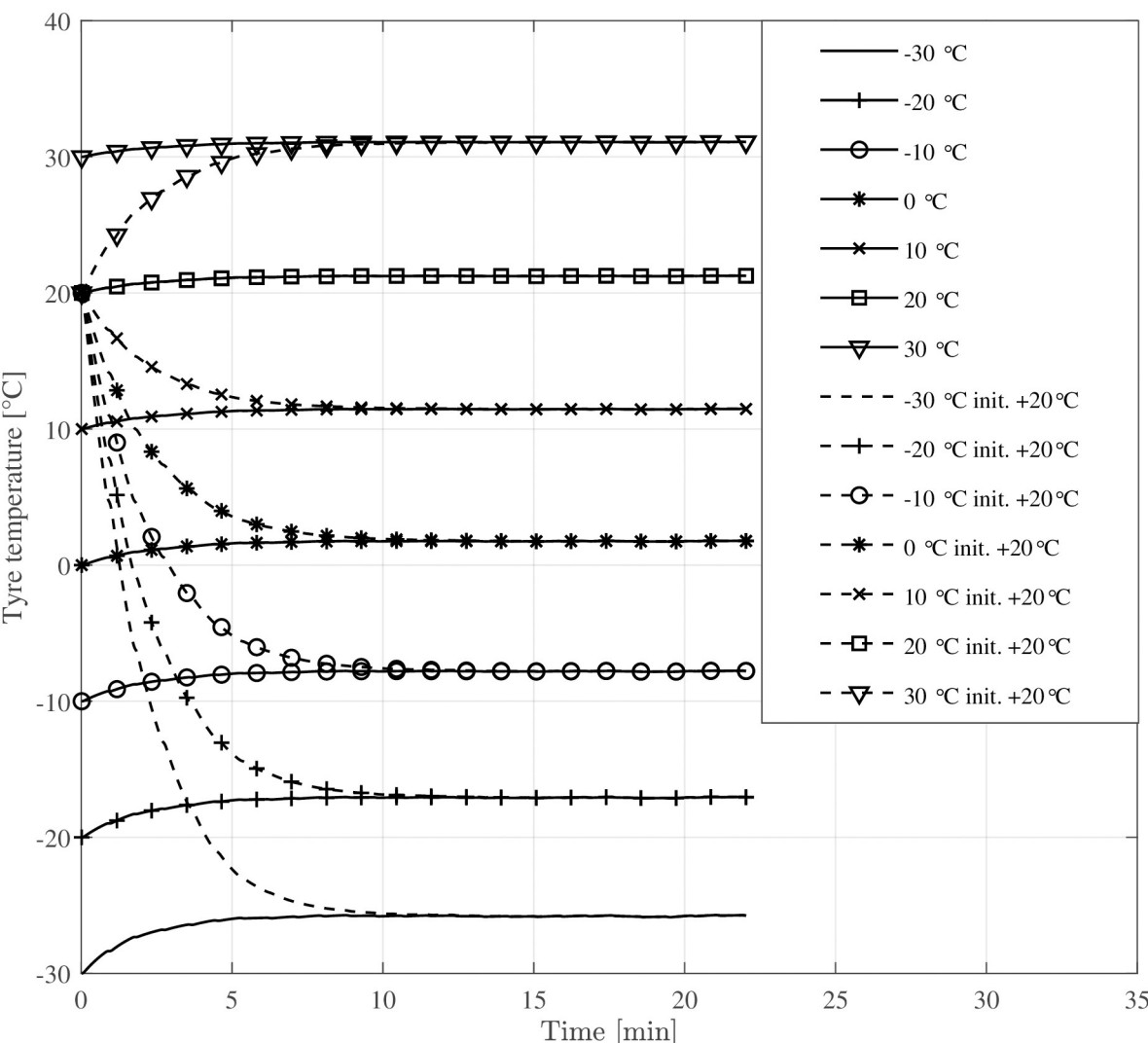

**Fig 8. Tyre temperatures during driving for the given drive cycle.** One set of simulations starting at ambient temperature (–) and one starting at + 20˚C (- -).

driving range drops significantly at decreasing ambient temperatures. Therefore, the lowest planned ambient temperature should be the dimensioning factor for the battery capacity.

## Discussion

Cycling during the winter season is a current topic that is accompanied by corresponding research as shown by the references in the introduction. Accordingly, numerous cities in northern countries have adopted winter cycling strategies in recent years, with primary focus on de-icing and snow ploughing.

The simulation results indicate that the temperature rise of bicycle tyres during operation is small compared to car or truck tyres. However, at low ambient temperatures (below the freezing point), even a small increase in tyre temperature (Fig 8) causes a substantial reduction in rolling resistance (Fig 9). This work extends the temperature-dependent rolling resistance model proposed by Rothhämel [21] to also consider transient effects within the tyre.

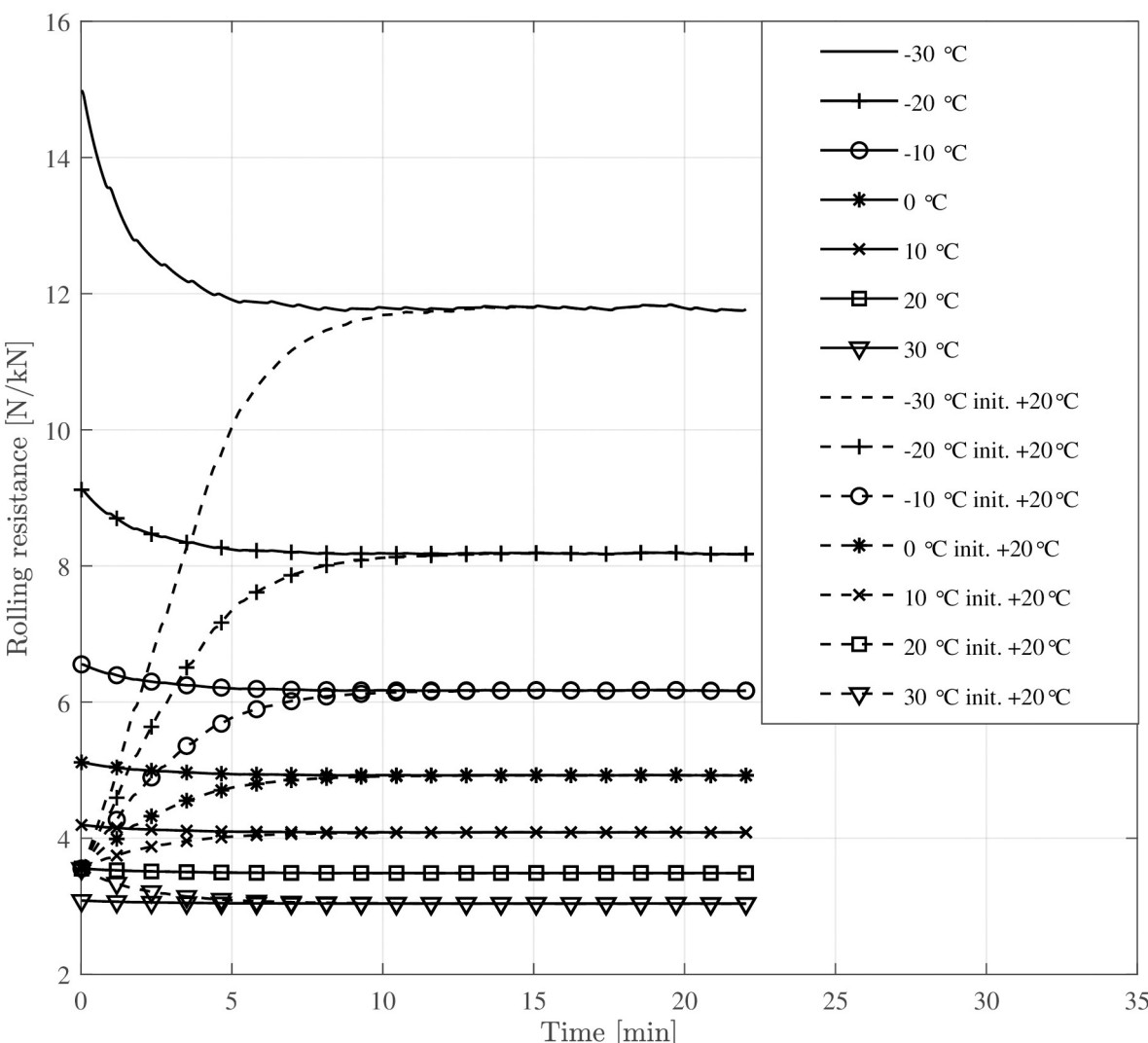

**Fig 9. Rolling resistance at different ambient temperatures for the given drive cycle.** One set of simulations starting at ambient temperature (–) and one starting at + 20˚C (- -).

The increase in tyre temperature during operation and the related decrease in rolling resistance depend on vehicle speed. The steady-state simulation (Figs 4 and 5) illustrates the transition very distinctly, even though it is not a realistic driving behaviour. A realistic driving cycle (Fig 7) based on statistical data was created to study commuting of cyclists. This kind of driving cycle is similar to those used for exhaust gas emission measurements in motor vehicles. The driving cycle facilitates simulation of the longitudinal dynamic behaviour of bicycles, enabling comparison of rolling resistance changes and its real-life implications. Although the driving cycle's input data may vary locally, including topographic effects, generating representative cycles is acknowledged as a distinct research field. The driving cycle is exchangeable for future applications of the present model if necessary. The model represents primarily bicycle traffic in built-up areas where most of the commuting takes place. This also includes frequent stops. In rural areas with a higher fraction of highway driving more steady-state cycling can be expected.

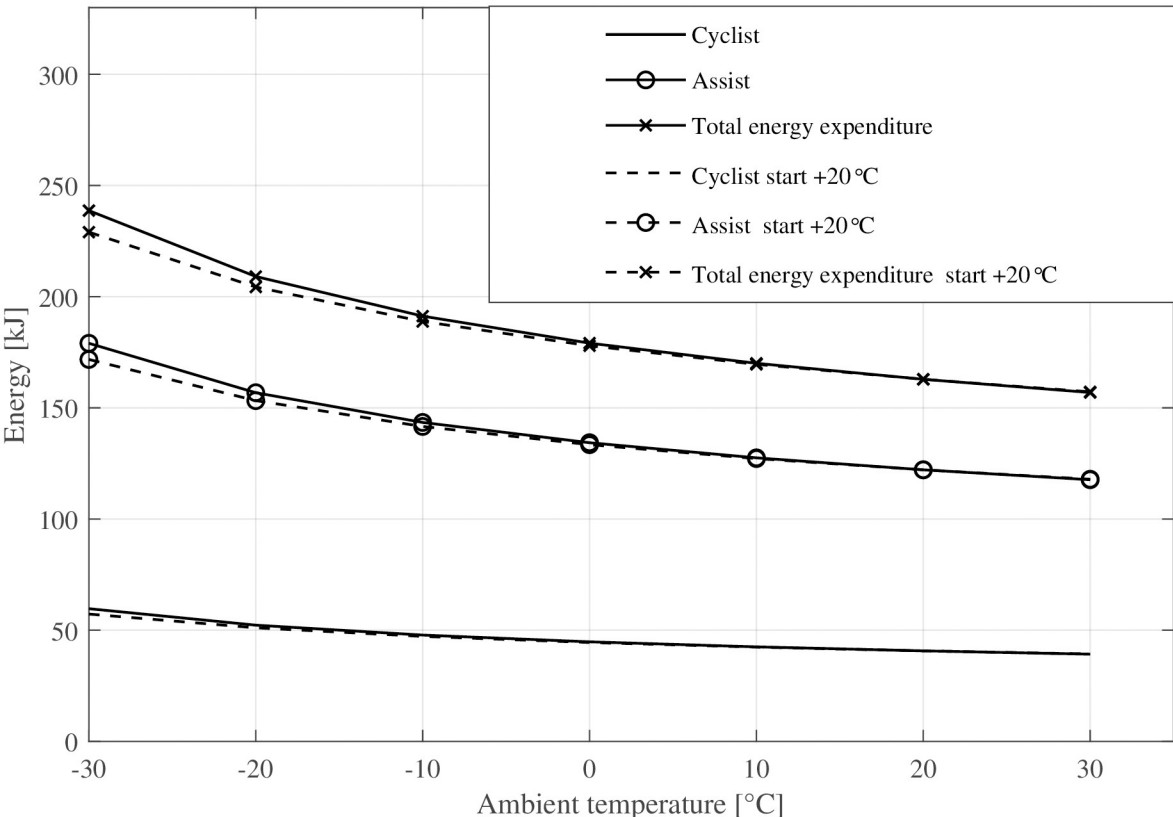

**Fig 10. The energy expenditure for the cyclist, electrical assistance, and the total energy expenditure over ambient temperature.**

The presented transient tyre temperature and rolling resistance model has a physical basis and it is tuned using prior measurements [21]. The model showcases changes in rolling resistance similar to those observed in truck tyres [31]. However, on a bicycle the transient effects take place on a smaller scale. This can be explained by the smaller masses and a lower amount of energy losses in the system. Even if the masses are smaller, the ratio is in a similar order of magnitude: the tyres of a bicycle and truck account for approximately 1–2% of the total mass.

The input data that were used to tune the model, originates from earlier measurements, covering a temperature range from −30°C to + 25°C. Cyclists might also operate their bicycles at higher ambient temperatures. However, the temperature dependency of rolling resistance diminishes with increasing ambient temperature, while air density decreases, causing a decrease in air drag. It can be assumed that the model is also valid up to a certain limit at higher ambient temperature than that covered by the original data. However, the largest limitation of the model at significantly higher temperatures is due to the absence of a human body efficiency model. Beyond a certain threshold, the human propulsion capacity starts to decrease significantly.

The simulations suggest that commuting durations less than 5–10 minutes will result in significantly lower rolling resistance, particularly at temperatures below + 10°C, if the bicycle has been stored in a warm garage.

## Conclusions and future work

This article presented a model that can capture the transient rolling resistance of a bicycle tyre at different ambient temperatures. The impact of ambient and tyre temperatures can be

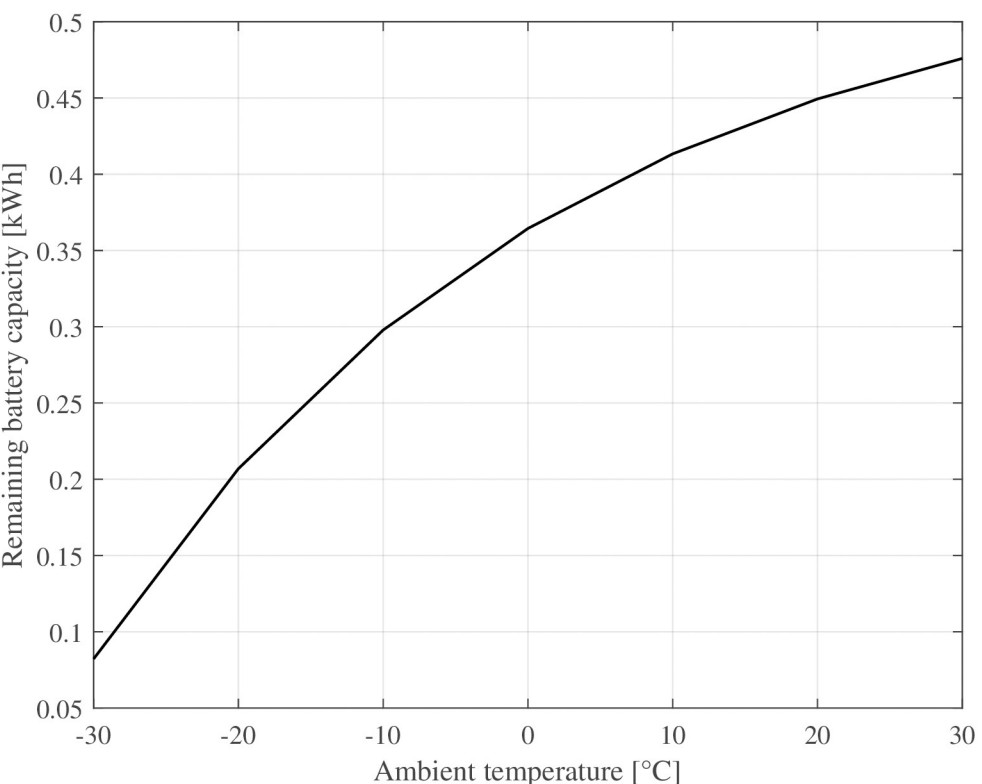

**Fig 11. The remaining battery capacity over ambient temperature for the given driving cycle.**

identified for a cyclist using the developed temperature-dependent bicycle rolling resistance model. Furthermore, this model was used to simulate the energy expenditure changes at various ambient temperatures for an electrically assisted bicycle. As a result, the following conclusions can be made:

- The developed rolling resistance model effectively captures the transient rolling resistance of bicycle tyres at various ambient temperatures and drive times. Consistent with prior investigations [31, 37], the general characteristic behaviour of warming-up and subsequent reduction in rolling resistance post-start is successfully reproduced.

- Even small increases in tyre temperature at cold ambient temperatures decrease the rolling resistance considerably. This is exemplified in Fig 5 where a change in riding speed from $30\,^{km}/_h$ to $10\,^{km}/_h$ (e. g. cycling from a slight declination to an up-hill section), leads to a decreased energy dissipation, resulting in a lower tyre temperature and approximately 10% higher rolling resistance. This change is equivalent to altering the inflation pressure from 400 kPa to 300 kPa.

- At warm temperatures, the warm-up behaviour of the tyre has only a minor effect (1–2%) on rolling resistance.

- The ambient and tyre temperatures greatly influence a bicycle's driving losses—i. e., rolling and aerodynamic resistance. This means that a lowered temperature causes an unfavourable combination of all parameters affecting the required propulsion energy. A drop of ambient temperature by + 10˚C results in a 10–15% increase of energy expenditure for a given driving cycle (see Fig 10).

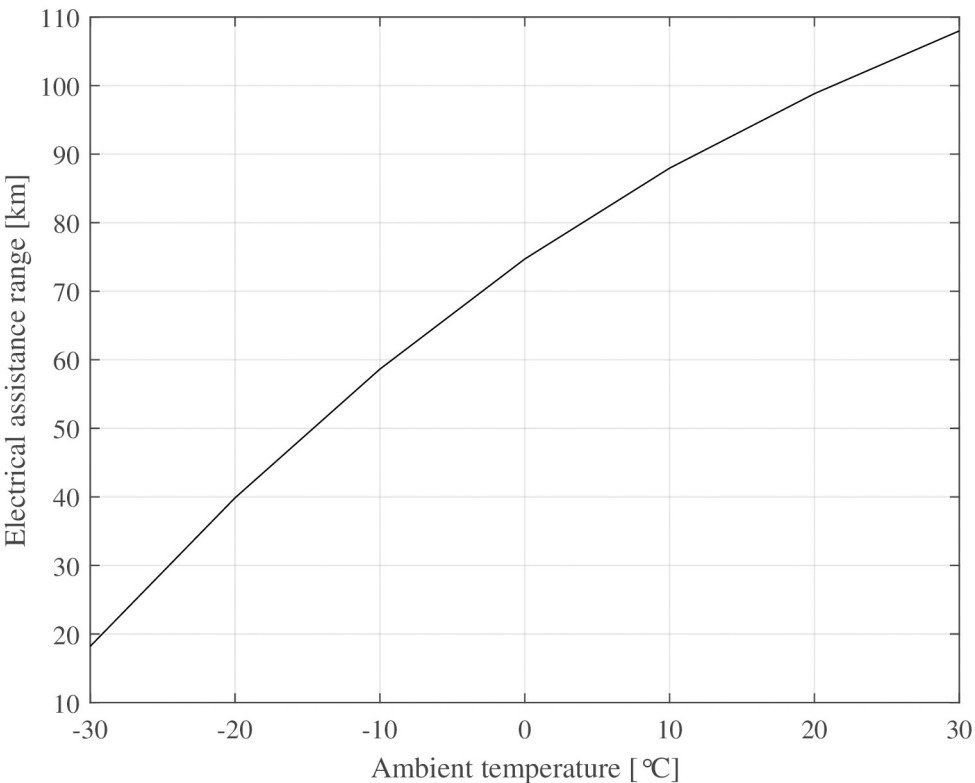

**Fig 12. Driving range with electric assistance over ambient temperature for the given drive cycle.**

- Electrically assisted bicycles have a considerably shorter driving range during winter at cold temperatures. This is caused by the increased driving losses, specifically rolling and aerodynamic resistance (see above) and the decreased battery capacity (see Eq (22) and Fig 11). The cycling range drops over 20% from + 20˚C to 0˚C. The decline of cycling range is even more drastic below the freezing point. −10˚C is a typical limit for Li-Ion-batteries that is recommended by the manufacturers, where only 60% of the nominal range is available for the simulated pedelec.

- The increased driving losses during cold weather might decrease the cycling motivation of cyclists if gain factors are not adjusted at cold temperatures to compensate for the increased driving losses.

- The lowest operational temperature that the pedelec is planned to be used at, should be a dimensioning factor regarding the battery size and range aspect. For already existing pedelecs clear communication might help to increase customers' understanding. Even a second battery, as is standard for some electric bicycles, could be an extension of existing bicycles in some circumstances.

In summary, the study provides valuable insights into the dynamics of winter cycling, offering implications for both research methodologies and practical applications, especially in built-up areas where commuting by bicycles takes place most commonly. The presented model's adaptability to different driving cycles enhances its utility across diverse cycling scenarios.

In the future, the model should be validated using test rig or on-road experiments. In addition, simulations could be expanded to consider the increase of rolling resistance due to road

conditions, such as mud and snow, and road roughness or softness. It is recommended to parameterise the model for different types of tyres, as the model is currently parameterised using a dataset of only one type of conventional tyre. However, during winter conditions, studded winter tyres are often used. It would be beneficial to measure studded or unstudded winter tyres for better range estimation of pedelecs during the winter season.

## Supporting information

**S1 Data.**
(TXT)

## Author Contributions

**Conceptualization:** Jukka Hyttinen, Malte Rothhämel.

**Data curation:** Jukka Hyttinen, Malte Rothhämel.

**Formal analysis:** Jukka Hyttinen, Malte Rothhämel.

**Funding acquisition:** Jenny Jerrelind, Lars Drugge.

**Investigation:** Jukka Hyttinen, Malte Rothhämel.

**Methodology:** Jukka Hyttinen, Malte Rothhämel.

**Project administration:** Jenny Jerrelind, Lars Drugge.

**Resources:** Jukka Hyttinen, Jenny Jerrelind, Lars Drugge.

**Software:** Jukka Hyttinen.

**Supervision:** Malte Rothhämel, Jenny Jerrelind, Lars Drugge.

**Validation:** Jukka Hyttinen, Malte Rothhämel.

**Visualization:** Jukka Hyttinen, Malte Rothhämel.

**Writing – original draft:** Jukka Hyttinen, Malte Rothhämel.

**Writing – review & editing:** Malte Rothhämel, Jenny Jerrelind, Lars Drugge.

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
