## [Decision Letter · Decision Letter 0]

24 Nov 2023

PONE-D-23-29914Simulation of non-stationary rolling resistance of bicycle tyres at various ambient temperaturesPLOS ONE

Dear Dr. Rothhämel,

Thank you for submitting your manuscript to PLOS ONE. After careful consideration, we feel that it has merit but does not fully meet PLOS ONE’s publication criteria as it currently stands. Therefore, we invite you to submit a revised version of the manuscript that addresses the points raised during the review process.

In addition to addressing the comments from the reviewers, it is important that the authors must highlight the relevance and significant novelty or innovation of this article for publication in a high-impact factor journal. 

We look forward to receiving your revised manuscript.

Kind regards,

A A Chowdhury, Ph.D., FHEA

Academic Editor

PLOS ONE

“The project belongs to the Centre for ECO2 Vehicle Design, which is funded by the Swedish Innovation Agency Vinnova (Grant Number 2016-05195). The authors wish to thank the Centre for ECO2 Vehicle Design, the strategic research area TRENoP and Scania for their financial support.”

“The project belongs to the Centre for ECO2 Vehicle Design, which is funded by the Swedish Innovation Agency Vinnova (Grant Number 2016-05195). The authors wish to thank the Centre for ECO2 Vehicle Design, the strategic research area TRENoP and Scania for their financial support.

Reviewers' comments:

**Comments to the Author**

1. Is the manuscript technically sound, and do the data support the conclusions?

Reviewer #1: Partly

Reviewer #2: Yes

Reviewer #3: Yes

Reviewer #4: Partly

2. Has the statistical analysis been performed appropriately and rigorously? 

Reviewer #1: N/A

Reviewer #2: Yes

Reviewer #3: Yes

Reviewer #4: N/A

3. Have the authors made all data underlying the findings in their manuscript fully available?

Reviewer #1: Yes

Reviewer #2: Yes

Reviewer #3: Yes

Reviewer #4: Yes

4. Is the manuscript presented in an intelligible fashion and written in standard English?

Reviewer #1: Yes

Reviewer #2: Yes

Reviewer #3: Yes

Reviewer #4: Yes

5. Review Comments to the Author

Reviewer #1: The referred paper which studied the stationary model of rolling resistance is not accessible to be able to compare the two studies and comment about the level of novelty of the work. Some sentences need to be revised.

Reviewer #2: The results are interesting but more suitable for a lower level of dissemination, such as press articles and books, rather than a high-impact factor journal. This is because the level of research in this study is quite basic, and it relies on several assumptions. The study does not bring significant novelty or innovation for publication in a journal of this caliber where it was submitted.

All of the conclusions are expected, and it's something that almost anyone is already aware of. For example, it is a well-known fact that battery capacity decreases with temperature. It is common knowledge that cold tires cannot deform as much as warm ones, leading to increased rolling resistance. We are well aware that air density increases with decreasing temperature. There are studies that show that predicted speeds are slightly lower when the temperature is lower, and the difference is about 0.2 km/h for a 5°C decrease. An important point to consider is that even if the article were of a high standard, its relevance would still be quite limited. This is because, in reality, the temperature range in which someone is cycling is much narrower. While the most significant variation was observed at -30°C and -20°C, one should ask how many people opt for cycling when they have a car, especially at these extreme temperatures.

The final reviewer's conclusion is that this study adds very little novelty to the topic it addresses.

Reviewer #3: The article is devoted to Modeling the unsteady rolling resistance of a bicycle tire at different ambient temperatures. The model calculates the tire temperature based on heat transfer taking into account heating: that is, rolling resistance and cooling effect, convective and radiative cooling. A decrease in tire temperature leads to an increase in rolling resistance and a decrease in battery capacity, which was taken into account in the modeling. However, there are some comments regarding the work:

1. The Abstract section must be rewritten to reflect the relevance of the problem being solved and the scientific novelty of the solution obtained.

2. Keywords must be adjusted, highlighting special terms that characterize the study.

3. At the end of the Introduction section, it is necessary to determine the relevance and purpose of the scientific review.

4. The labels in Figure 1 should be made clearer as it is blurry.

5. The list of cited sources should include more modern publications on the energy of transport and technical systems, for example,

https://doi.org/10.3390/app10082879

https://doi.org/10.3390/math11102394

6. Figure 6 shows a graph of battery capacity changes. How can we explain the increase in capacity above the nominal value at fairly high temperatures near 60 degrees Celsius?

7. In the Discussion section, it is necessary to characterize the obtained models and the scientific results obtained, describe their advantages and disadvantages, and also give the limitations of the proposed model of rolling resistance at different ambient temperatures.

8. Conclusions must be structured, highlighting the main scientific and practical results obtained, supporting them with numerical results.

Reviewer #4: The concept of simulating non-stationary rolling resistance in electrically assisted bicycles is both important and innovative because it addresses a practical issue that is experienced by urban commuters. It is praiseworthy that a model was developed to simulate the resistance of tyres as a function of the temperature of the surrounding environment. It is beneficial, from a practical standpoint, for producers of electric bicycles as well as users of these bicycles to highlight the impact of ambient temperature on the capacity of the battery and the rolling resistance. Obtaining a complete picture of the elements that influence tyre temperature requires taking into account not just the ones that cause heating, such as rolling resistance, but also those that cause cooling, such as convective and radiative impacts.

Areas for Improvement:

A more in-depth discussion of the model, particularly how it models the dynamic changes in tyre resistance and battery capacity, would be beneficial to include in this article.

Although the results of the simulation are interesting and useful, the article would be improved if it included empirical validation or testing in the real world to validate the model's predictions.

It would be helpful to have a conversation about the significance of these findings for the design of bicycles in the future as well as how they relate to already available bicycles.

The essay needs to discuss the limitations of the model and suggest other avenues for research, which could involve varying types of weather or tyres, for example.

The article might be more user-focused if it contained a part that discussed how commuters can directly benefit from this research. This section could include suggestions for temperatures that are best for commuting or techniques to avoid the effects of cold weather.

It is possible that the comprehension of the findings, as well as their accessibility, could be improved by the addition of graphical representations of the simulation results.

The article offers a substantial contribution to our comprehension of how the surrounding temperature influences the performance of electrically assisted bicycles. By addressing these technical concerns, the paper could enhance and apply its findings, making it more valuable for academics and industry practitioners.

6. PLOS authors have the option to publish the peer review history of their article (what does this mean?). If published, this will include your full peer review and any attached files.

Reviewer #1: **Yes: **Fatemeh Bahmani

Reviewer #2: No

Reviewer #3: **Yes: **Nikita V. Martyushev

Reviewer #4: No

---

## [Author Response · Author response to Decision Letter 0]

17 Jan 2024

To the Editor and Reviewers of PlosONE 

This information is given as a Rebuttal Letter, attached as PDF.

However, according to the form the information is given here, too.

Revised research paper (PONE-D-23-29914)

Dear Sir or Madam,

Thank you for your kind answer and the valuable feedback.

To reviewers: Thank you very much for your review comments on our paper. We experienced that the paper increased in quality when working on the suggested improvements. In this document we will go through your comments (in red) and refer to corresponding changes in the paper.

Editor’s comments:

Answer: The article has been modified to use Latex template.

Answer: We were not aware of that impact of depositing our data. Therefore, we will provide our driving cycle as a supplementary material. All of the other required information to reproduce the results is included in the parameters and equations.

“The project belongs to the Centre for ECO2 Vehicle Design, which is funded by the Swedish Innovation Agency Vinnova (Grant Number 2016-05195). The authors wish to thank the Centre for ECO2 Vehicle Design, the strategic research area TRENoP and Scania for their financial support.”

“The project belongs to the Centre for ECO2 Vehicle Design, which is funded by the Swedish Innovation Agency Vinnova (Grant Number 2016-05195). The authors wish to thank the Centre for ECO2 Vehicle Design, the strategic research area TRENoP and Scania for their financial support.

Answer: The funding information is now removed from the article and will be provided in the online form.

Answer: We will provide our driving cycle as a supplementary material. All of the other required information to reproduce the results is included in the parameters and equations.

Reviewers’ comments:

1) (Original comment) Grammar.

Answer 1): Grammar and typos have been addressed (see changes in document). 

2) (Original comment) In the last line of the abstract, please specify a range for very low ambient temperatures.

Answer: A good comment, thank you. It is updated in the abstract.

3) Line 41: Is the gain factor a parameter? If so, you may define it.

Answer: The gain factor can be a parameter and different bicycles can have different amounts of gains. For simplicity a gain factor of 3 was chosen in this case.

4) Line 44: please explain what you mean by range anxiety and whether it was meant to be pedelec or electric cars.

Answer: Range anxiety is now explained in the text (introduction).

5) Line 66: Which factor do you think has the most significant effect on the rolling resistance among the factors mentioned? Does ambient temperature have the highest impact?

Answer: At temperatures above zero Celsius, the tyre inflation pressure is the most important factor for rolling resistance. However, for temperatures below zero degrees, the influence of tyre temperature is higher than the tyre inflation pressure on rolling resistance. These points have been added in the introduction.

6) Rolling resistance force is mentioned as Fr in equation (1) but labeled as Frr in Figure 1

Answer: Thank you for a good comment. It is changed now to Fr.

7) If this is Newton’s second law, should not Ftotal be equal to zero? Where was this derivation applied to create the results?

Answer: Ftotal is the total driving losses. This could be reformulated to second Newton’s law, but in this case describes the losses that must be overcome.

8) Line 101: You may replace weight with mass.

Answer: It is replaced now.

9) Please add units for all the parameters and variables in the equations.

Answer: Units and variables are shown in Table 1.

10) Fc in figure 2 is not defined.

Answer: Figure is modified now so that contact reaction force is changed to Fz.

11) Small v is used in equation (3) and capital V in equation (6) for velocity. You may use the same notation for the variables.

Answer: Equation 6 is modified so that consistent notation is used for the velocity.

12) Line 155: I assume it was meant to be “rolling resistance coefficient”.

Answer: In this case it was meant to be rolling resistance force, which is also updated in the text.

13) Define b1 and b2 in equation (12).

Answer: The regression parameters were taken from another paper and are now also explained better in the text.

14) Equation (17) is not clear, hfree which is mentioned in the text is not used in this equation and hnatural is not defined. 

Answer: Now only hnatural is used in the article.

15) Where is the non-stationary assumption implemented in the derivations compared to the referred steady-state study?

Answer: The transient rolling resistance is simulated in this study. In contrast, usual studies assume only constant rolling resistance value. Non-stationary word has been replaced by transient in the text.

16) Unit of the vertical axis in figure 6 is missing.

Answer: The figure has been modified and units are added.

17) Line 281: The sentence is incomplete.

Answer: The sentence has been modified.

Reviewer #1: The referred paper which studied the stationary model of rolling resistance is not accessible to be able to compare the two studies and comment about the level of novelty of the work. Some sentences need to be revised.

Answer: Thank you for your comments. We assume that your comment refers to the article “On rolling resistance of bicycle tyres with ambient temperature in focus”, which is accessible now with open access.

The paper is now updated with all comments taken into consideration. Larger changes are marked in blue.

Reviewer #2: The results are interesting but more suitable for a lower level of dissemination, such as press articles and books, rather than a high-impact factor journal. This is because the level of research in this study is quite basic, and it relies on several assumptions. The study does not bring significant novelty or innovation for publication in a journal of this caliber where it was submitted.

All of the conclusions are expected, and it's something that almost anyone is already aware of. For example, it is a well-known fact that battery capacity decreases with temperature. It is common knowledge that cold tires cannot deform as much as warm ones, leading to increased rolling resistance. We are well aware that air density increases with decreasing temperature. There are studies that show that predicted speeds are slightly lower when the temperature is lower, and the difference is about 0.2 km/h for a 5°C decrease. An important point to consider is that even if the article were of a high standard, its relevance would still be quite limited. This is because, in reality, the temperature range in which someone is cycling is much narrower. While the most significant variation was observed at -30°C and -20°C, one should ask how many people opt for cycling when they have a car, especially at these extreme temperatures.

The final reviewer's conclusion is that this study adds very little novelty to the topic it addresses.

Answer: Thank you for your comments. However, we respectfully disagree: The increase in rolling resistance at cold temperatures is not because the tyres deform less at lower ambient temperature. It is increased damping of material at cold temperatures that causes the increase in rolling resistance. 

We have tried to find similar articles regarding bicycle tyre models with transient temperature dependent behaviour, however, we did not find any. We would be happy if a reference to a similar article could be provided. Regarding temperature range of cycling, in Nordic countries it is not uncommon to cycle even at extremely cold temperatures (e.g. in Stockholm or Helsinki you still see cyclists on the road at -25°C). To clarify this, we have added two sentences in the introduction including references and in the discussion section. In this context, these types of models can be used in the design process of electric bicycles to quantify what kind of battery capacities are suitable for the climate that the bicycle is supposed to be used at, or to show the rider how much range there is left at current ambient temperature. To further clarify this, we modified figures 8-10 to include also simulations when bicycle has been kept in a warm garage and tyres are initially warm. 

In addition, traffic planners in municipalities are requesting for simulation tools that take this kind of parameters into account to increase convenience of cycling to reduce motorised traffic. According to our knowledge, these people are not only interested in the summarising description of decrease speed based on temperature, but also in the reasons for understanding how the infrastructure can be changed for more environmentally friendly solutions. We are convinced that our model contributes to this understanding.

Reviewer #3: The article is devoted to Modeling the unsteady rolling resistance of a bicycle tire at different ambient temperatures. The model calculates the tire temperature based on heat transfer taking into account heating: that is, rolling resistance and cooling effect, convective and radiative cooling. A decrease in tire temperature leads to an increase in rolling resistance and a decrease in battery capacity, which was taken into account in the modeling. However, there are some comments regarding the work:

1. The Abstract section must be rewritten to reflect the relevance of the problem being solved and the scientific novelty of the solution obtained.

Answer: Thank you for your comments. The abstract was modified to emphasize the novelty and relevance of addressed aspects.

2. Keywords must be adjusted, highlighting special terms that characterize the study.

Answer: We have updated the keywords 

3. At the end of the Introduction section, it is necessary to determine the relevance and purpose of the scientific review.

Answer: The end of the introduction was changed. A next-to-last paragraph was added to motivate how the results of our study can be used.

4. The labels in Figure 1 should be made clearer as it is blurry.

Answer: The figure is a vector picture that is scalable without quality loss. If the picture is blurry, it might be because of the compression in the review system. It will be a higher quality picture when the paper is published. We have changed the article template to Latex template, which should solve the problem.

5. The list of cited sources should include more modern publications on the energy of transport and technical systems, for example,

https://doi.org/10.3390/app10082879

https://doi.org/10.3390/math11102394

Answer: Thank you for the suggestion. We have clarified the background aspects regarding energy and greenhouse gas emissions in the introduction. To support this, we have added to the manuscript two relevant modern publications (2020 and 2015), specific for electrified bicycles. 

6. Figure 6 shows a graph of battery capacity changes. How can we explain the increase in capacity above the nominal value at fairly high temperatures near 60 degrees Celsius?

Answer: The change in battery capacity is due to changes in molecular movements. Higher temperatures increase molecular movement while lower temperatures decrease it. The nominal capacity is merely a normalised value of battery capacity at room temperature. Therefore, over this temperature, the battery capacity can increase over the nominal value.

7. In the Discussion section, it is necessary to characterize the obtained models and the scientific results obtained, describe their advantages and disadvantages, and also give the limitations of the proposed model of rolling resistance at different ambient temperatures.

Answer: A discussion section has been added to address these issues.

8. Conclusions must be structured, highlighting the main scientific and practical results obtained, supporting them with numerical results.

Answer: The Conclusions sections has been complemented as recommended with numbers and internal references.

Reviewer #4: The concept of simulating non-stationary rolling resistance in electrically assisted bicycles is both important and innovative because it addresses a practical issue that is experienced by urban commuters. It is praiseworthy that a model was developed to simulate the resistance of tyres as a function of the temperature of the surrounding environment. It is beneficial, from a practical standpoint, for producers of electric bicycles as well as users of these bicycles to highlight the impact of ambient temperature on the capacity of the battery and the rolling resistance. Obtaining a complete picture of the elements that influence tyre temperature requires taking into account not just the ones that cause heating, such as rolling resistance, but also those that cause cooling, such as convective and radiative impacts.

Areas for Improvement:

A more

---

## [Decision Letter · Decision Letter 1]

7 Feb 2024

PONE-D-23-29914R1Simulation of transient rolling resistance of bicycle tyres at various ambient temperaturesPLOS ONE

Dear Dr. Rothhämel,

Thank you for submitting your manuscript to PLOS ONE. After careful consideration, we feel that it has merit but does not fully meet PLOS ONE’s publication criteria as it currently stands. Therefore, we invite you to submit a revised version of the manuscript that addresses the points raised during the review process.

Kind regards,

Ashfaque Chowdhury, Ph.D., FHEA, FIEB

Academic Editor

PLOS ONE

Journal Requirements:

**Comments to the Author**

1. If the authors have adequately addressed your comments raised in a previous round of review and you feel that this manuscript is now acceptable for publication, you may indicate that here to bypass the “Comments to the Author” section, enter your conflict of interest statement in the “Confidential to Editor” section, and submit your "Accept" recommendation.

Reviewer #1: (No Response)

Reviewer #3: All comments have been addressed

Reviewer #4: All comments have been addressed

2. Is the manuscript technically sound, and do the data support the conclusions?

Reviewer #1: Yes

Reviewer #3: Yes

Reviewer #4: Yes

3. Has the statistical analysis been performed appropriately and rigorously? 

Reviewer #1: Yes

Reviewer #3: Yes

Reviewer #4: Yes

4. Have the authors made all data underlying the findings in their manuscript fully available?

Reviewer #1: Yes

Reviewer #3: Yes

Reviewer #4: Yes

5. Is the manuscript presented in an intelligible fashion and written in standard English?

Reviewer #1: No

Reviewer #3: Yes

Reviewer #4: Yes

6. Review Comments to the Author

Reviewer #1: I am not convinced about the gain factor and still think it should be properly defined.

Also the English needs to be impoved.

Reviewer #3: In general, the authors have given an answer to my comments. And the necessary corrections were made in the text of the article.

The authors' response to my comment No. 6 should be added to the description of Figure 6 in the text of the article.

6. Figure 6 shows a graph of battery capacity changes. How can we explain the

increase in capacity above the nominal value at fairly high temperatures near 60

degrees Celsius?

Answer: The change in battery capacity is due to changes in molecular movements.

Higher temperatures increase molecular movement while lower temperatures decrease

it. The nominal capacity is merely a normalised value of battery capacity at room

temperature. Therefore, over this temperature, the battery capacity can increase over

the nominal value.

Reviewer #4: The authors properly responded to the comment given by the reviewers, as well as the manuscript has been modified accordingly. Now, this manuscript can be accepted for publication.

7. PLOS authors have the option to publish the peer review history of their article (what does this mean?). If published, this will include your full peer review and any attached files.

Reviewer #1: No

Reviewer #3: No

Reviewer #4: No

---

## [Author Response · Author response to Decision Letter 1]

19 Mar 2024

Dear Sir or Madam,

Thank you for your kind answer and the valuable feedback.

To reviewers: Thank you very much for your review comments on our paper. The paper has now undergone professional language editing (comment from reviewer #1). In addition, the gain factor was defined in section “Electrically assisted bicycles and energy consumption” (comment from reviewer #1). Moreover, we have adapted the text around Fig. 6 according to our former explanation regarding the capacity of the battery and its possibility to exceed 100% (comment from reviewer #3). All changes are marked in blue. We hope that the paper now fulfils all requirements for publication.

Sincerely,

Malte Rothhämel

---

## [Decision Letter · Decision Letter 2]

15 Apr 2024

Simulation of transient rolling resistance of bicycle tyres at various ambient temperatures

PONE-D-23-29914R2

Dear Dr. Rothhämel,

We’re pleased to inform you that your manuscript has been judged scientifically suitable for publication and will be formally accepted for publication once it meets all outstanding technical requirements.

Kind regards,

Ashfaque Ahmed Chowdhury, Ph.D., FHEA, FIEB

Academic Editor

PLOS ONE

---

## [Editor Report · Acceptance letter]

17 Jun 2024

PONE-D-23-29914R2 

PLOS ONE

Dear Dr. Rothhämel, 

I'm pleased to inform you that your manuscript has been deemed suitable for publication in PLOS ONE. Congratulations! Your manuscript is now being handed over to our production team.

Kind regards, 

on behalf of

Dr. Ashfaque Ahmed Chowdhury 

Academic Editor

PLOS ONE